# Biomarkers and Fever in Children with Cancer: Kinetics and Levels According to Final Diagnosis

**DOI:** 10.3390/children8111027

**Published:** 2021-11-09

**Authors:** Ana de Lucio Delgado, Jose Antonio Villegas Rubio, Corsino Rey Galan, Belen Prieto García, Maria de los Reyes González Expósito, Gonzalo Solís Sánchez

**Affiliations:** 1Pediatric Department, Hospital Universitario Central de Asturias, 33011 Oviedo, Spain; javillegas10@hotmail.com; 2Pediatric Intensive Care, Hospital Universitario Central de Asturias, 33011 Oviedo, Spain; corsino.rey@gmail.com; 3Clinical Biochemistry, Laboratory of Medicine, Hospital Universitario Central de Asturias, 33011 Oviedo, Spain; prietobelen7@gmail.com; 4Institute of Cancer Research, London SW3 6JB, UK; reyes.gonzalez.exposito@gmail.com; 5Service of the Neonatology Unit, Hospital Universitario Central de Asturias, 33011 Oviedo, Spain; gonzalosolissanchez7@gmail.com

**Keywords:** cancer, children, fever

## Abstract

We investigated the kinetics of CRP, PCT, IL-6 and MR-proADM in a cohort of consecutive febrile patients with cancer in order to test the hypothesis that higher plasma concentrations and the absence of a rapid decrease in peak values would be associated with disease severity. (1) Method: A prospective descriptive and analytical study of patients with cancer and fever (≤18 years of age) at a University Hospital was carried out between January 2018 and December 2019. Information collected: sex, age, diagnosis, date and symptoms at diagnosis and medical history. The episodes were classified into three groups: bacterial infection, non-bacterial infection and systemic inflammatory response syndrome (SIRS). (2) Results: One hundred and thirty-four episodes were included. Bacterial infection criteria were met in 38 episodes. Biomarkers were measured at four different points: baseline, at 12–24 h, at 25–48 h and at 49–72 h. All the biomarkers evaluated decreased after the peak level was reached. IL-6 and MR-proADM showed a trend towards higher levels in the SIRS group although this rise was statistically significant only for IL-6 (*p* < 0.005). Bacterial infections more frequently presented values of PCT above the cut-off point (>0.5 ng/mL) at 12–24 h. (3) Conclusion: In our experience, IL-6 kinetics is faster than PCT kinetics and both are faster than CRP in patients with fever and cancer who present a good outcome. Patients with a good evolution show a rapid increase and decrease of PCT and particularly of IL-6 levels.

## 1. Background

Childhood cancer has a significant social and health impact. According to data based on the population coverage areas of the National Registry of Childhood Tumors, the crude incidence rate of childhood cancer in Spain for the 2000–2015 period was 157.0 cases per million children aged 0–14 years [1]. The cases of tumors in pediatric age diagnosed in Spain correspond to a Western, industrial and developed country, very similar to other European countries.

Childhood cancer survival rates have improved in recent decades. Nowadays, approximately 80–85% of patients survive beyond 5years from the time of diagnosis. Advances in therapeutic strategies and supportive care have contributed to this improvement in the survival rate [2].

Frequently, children with undiagnosed cancer come to Emergency Departments (ED) with shortly after the onset of fever and neutropenia. Obtaining blood biomarkers to determine the disease prognosis constitutes a clinical necessity. Different tools have been evaluated to establish an early diagnosis of bacterial infection through the use of inflammatory response biomarkers such us C-reactive protein (CRP), procalcitonin (PCT), interleukin 6 (IL-6) and more recently, mid-regional pro-adrenomedullin (MR-proADM).

CRP is the most widely used biomarker. CRP reaches its maximum peak 36–48 h after the onset of the infectious episode. The slow kinetics showed by this biomarker can be the reason for false negative results at this time point of the episode [3,4]. PCT has been identified as a marker of bacterial sepsis and serious infections with a higher diagnostic accuracy than CRP [5,6,7]. Due to its favorable kinetics, it is considered more adequate for use in the ED, since it rises only 2–6 h after the bacterial stimulus and the maximum values will be detected at 12–36 h. Regarding IL-6, its peak value is reached 2–3 h after the onset of fever, returning to basal levels at 6–8 h [8]. MR-proADM is produced and secreted by multiple mammalian tissues during physiological and infectious stress [9]. In recent years, its concentrations were shown to increase gradually in correlation with the severity of the disease [10,11]. There are a few studies that have analyzed the kinetics of CRP, PCT and IL-6 in pediatric cancer patients with fever [12]. However, to our knowledge, the kinetics of MR-proADM has not been studied in this group of patients at the same time as the other three biomarkers.

In this study, we investigated the kinetics of CRP, PCT, IL-6 and MR-proADM in a well-defined cohort of consecutive febrile patients with cancer in order to test the hypothesis that higher plasma concentrations and the absence of a rapid decrease in peak values would be associated with disease severity.

## 2. Materials and Methods

### 2.1. Study Design and Patients

In this prospective descriptive and analytical study, we evaluated febrile episodes in pediatric patients with cancer between January 2018 and December 2019 at a University Hospital. Patients aged 18 years or younger who were febrile at the time of admission to the ED or were already in the hospital ward were enrolled. We excluded patients with non-oncological disease or no data collection during the first 24 h of the episode. All episodes included in the study occurred in patients receiving chemotherapy treatment at the time of the febrile episode or had received it within the 3previous months. The recruitment period for patients who had an allogeneic bone marrow transplant and were still receiving immunosuppressive treatment was twelve months after chemotherapy started.

Next, we collected data pertaining to admission date and diagnosis, medical history including underlying cancer characteristics, disease staging and comorbidities. The first temperature obtained in the hospital and the temperature measured at home were both recorded. The duration of the fever and associated clinical symptoms, if any, were also collected. Furthermore, patients were classified microbiologically if a documented infection (bacterial, fungal or viral) was present.

Approval from the Autonomic Ethics Committee of Hospital Universitario Central de Asturias (Ethics Committee approval number: 109/15) was obtained. Written informed consent was signed by the patients’ parents or guardians and by children aged 16 years or older.

### 2.2. Diagnostic Groups

The episodes were grouped into three exclusive and broad groups: bacterial infection (BI), non-bacterial infection (NBI) including fever of unknown origin (FUO) or viral or local fungal infections and systemic inflammatory response syndrome (SIRS).

### 2.3. Definitions

Fever was defined as a central body temperature ≥38 °C or low-grade fever maintained ≥37.5 °C at least during an h plus impact on general condition(checked by the hospital doctor). A bacterial infection was defined as documented microbiological evidence of any bacteria in any sample obtained. Bacteremia was defined as the presence of bacteria in blood in peripheral culture or peripheral plus central culture. Catheter infection was defined as the presence of bacteria in blood in central culture. Sepsis was defined by the presence of both a culture-proven infection and SIRS. Patients with fever in whom all diagnostic tests were negative for bacterial, viral and fungal infections were considered to have no evidence of infection and were categorized as FUO. SIRS was defined on the basis of the consensus panel of the International Consensus Conference on Pediatric Sepsis [13]. Short-term evolution was defined as: (1) Patients located at home: need for hospital admission vs. home discharge. (2) Patients located at the hospital: clinical evolution within the first 72 h of the episode (disappearance of fever and no need to change antibiotic was considered a favorable outcome). A comorbidity was defined as: previous fever and neutropenia and/or previous fungal infections and/or port-a-cath infection in the last 6months, and/or endocrinological abnormalities and/or loss of weight of more than 10%. Finally, an uncontrolled disease was considered as a relapse and/or progression disease.

### 2.4. Interpretation of Biomarker Values

In our study, previous published biomarker cutoff values for BI were considered as reference values (CRP > 4 mg/Dl [14], PCT > 0.5 ng/mL [15], IL-6 > 85 pg/mL (abnormal range described between 50 pg/mL to 200 pg/mL) [16], MR-proADM > 0.5 nmol/L [17]). We analyzed the number of patients with biomarker levels above the suggested cutoff values at the moment that the biomarker reached the highest value. A reference of baseline median values of biomarkers was obtained from a group of non-oncological patients with no fever conditions admitted in our Hospital (CRP < 0.1 mg/dL, PCT< 0.1 ng/mL and IL-6 < 30 pg/mL) [18].

### 2.5. Laboratory Methods

An analytical control (blood count), biochemistry with biomarkers determination, peripheral/central blood cultures, urine (systematic and sediment and culture) and pharyngeal exudate for viruses and bacteria were collected within 12 h from the onset of fever. Residual samples were obtained at 12–24 h, 25–48 h and 49–72h after fever commenced. Blood samples were drawn into tubes containing lithium-heparin as anticoagulant for determination of CRP, PCT and IL-6. An additional tube containing plasma EDTA was also drawn for freezing at −80 °C and further processing of MR-proADM. Plasma CRP was measured in a Cobas 8000/module C501, PCT and IL-6 were measured in a Cobas E601 (Roche Diagnostic, Mannheim, Germany) whereas MR-proADM was measured in a Kryptor^®^ (Thermo Fisher Scientific, Hennigsdorf, Germany). Analytical detection limits were 0.07 mg/dL for CRP, 0.02 ng/mL for PCT, 1.5 pg/mL for IL-6 and 0.08 nmol/L for MR-proADM.

### 2.6. Statistical Analysis

Patients’ clinical and biological parameters were described using mean, 95% confidence interval of the mean (95% CI), median and interquartile range (IQR). Normal distribution was verified by Kolmogorov–Smirnov test, *p* < 0.05). Qualitative variables were analyzed by frequency distribution. The Student’s t-test was used for variables with normal distribution, and the Mann–Whitney U test and Kruskal–Wallis non parametric tests, in case an assumption of normality was not fulfilled. Pearson’s Chi-square test or Fisher’s exact test were used for establishing associations between qualitative variables. The program used was IBM SPSS statistics v24.

## 3. Results

### 3.1. Descriptive Study: Epidemiological Data

One hundred and thirty-four episodes in 37 patients were selected for our cohort. Patients with non-oncological hematological diseases (medullary aplasia, for example) from whom a sample of biomarkers had been collected, were excluded for data analysis. Those from whom the first blood sample was collected beyond the first 24 h after starting the febrile episode were also excluded. In total, the number of discarded episodes was 10(seven patients). Baseline demographic, clinical and laboratory characteristics of the patients are shown in Table 1. No significant differences were found between age at the time of the episode and sex (median of 8.2 years for males and 7.1 for females; *p* = 0.285). Furthermore, no significant association was observed between sex/tumor type of the patients (six males and 12 females for acute lymphoblastic leukemia (ALL)/acute myeloblastic leukemia (AML)/non-Hodgkin’s lymphoma (NHL) and eight men and 11 women for solid tumor/Hodgkin’s lymphoma (HL) or for age/tumor type (mean age of 7.7 years for ALL/AML/NHL and 6.6 years for solid tumor/HL) (*p* = 0.582 and *p* = 0.502, respectively).

The criteria for BI were met in 38 (28.3%) episodes: 15 Gram negative bacilli (GNB), 22 Gram positive bacilli (GPB) and a case of pneumonia (BI for clinical and radiological criteria without microbiological isolation). Eighty-eight episodes (65.6%) had fever but did not meet either SIRS or BI criteria. In this group, there were 40 episodes with documented germs (45.4%) including two fungal infections (2.2%) and 38 viral infections (43.1%) while no germs were found in 48 (54.4%). Finally, eight episodes (5.9%) were diagnosed with SIRS. Table 2 shows the different diagnoses and the location of infections.

### 3.2. Biomarkers and Final Diagnosis

(A) Kinetics of biomarkers.

Biomarkers were measured at four different time points: baseline, at 12–24 h, at 25–48 h and at 49–72 h. Figure 1 and Table 3 show the biomarkers kinetics for all episodes in relation with the final diagnosis. Mean CRP and PCT peak levels were reached at 25–48 h, and 12–24 h respectively, whereas the mean IL-6 peak level was observed earlier, at the baseline time point, except for the group of BI that reached the peak level at 12–24 h. Finally, the mean MR-proADM peak level was also detected at baseline, except for the group of SIRS that reached the peak level at 12–24 h. All the biomarkers evaluated decreased progressively after the peak level was reached.

We wanted to relate the value of biomarkers and the neutrophils count, not finding statistical significance for any of them (Table 4).

(B) Levels of biomarkers according to the final diagnosis.

When BI, NBI and SIRS were compared at the baseline control, the values of PCT, IL-6 and MR-proADM showed a trend towards higher levels in SIRS group, although this elevation was statistically significant only for IL-6 (SIRS with respect to NBI, *p* < 0.005). These results were also observed in the second analytical control and IL-6 was this time significantly higher in BI group (BI with respect to NBI and SIRS, *p* < 0.005) (Table 3). After excluding the eight episodes of SIRS, biomarkers values were again compared with the final diagnosis in the baseline analytical as well as at 12–24 h. The IL-6 level were significantly higher in both determinations for the BI group (*p* < 0.005).

The value of PCT > 0.5 ng/mL was statistically significant for SIRS with respect to BI and NBI. The value of IL-6 < 85 pg/mL was more frequent in NBI (*p* < 0.007). Episodes of BI were compared to episodes of NBI after excluding SIRS episodes. Only values above the cut-off point at 12–24 h for PCT were observed more frequently in BI episodes (*p* < 0.005). None of the other three analytical controls performed showed a statistically significant association between the four biomarkers and the final diagnosis (Table 5).

There was a correlation between all biomarkers except for CRP and MR-proADM. The largest linear regression observed was for MR-proADM in relation to PCT (R^2^ = 0.943), as it is showed in Table 6.

## 4. Discussion

### 4.1. Biomarker Levels and Kinetics

Our main objective was to analyze the usefulness of four different biomarkers (CRP, PCT, IL-6 and MR-proADM) as outcome predictors in febrile pediatric patients with cancer. To this end, we were able to describe the kinetics of these four biomarkers in these groups of patients. In our work, all patients showed a favorable outcome. There are few studies that analyzed together the kinetics of CRP, PCT and IL-6 in pediatric patients with fever and neutropenia [19] and to our knowledge this is the first study that describes MR-proADM kinetics with the rest of them. Our results confirmed that the kinetics of these four biomarkers is similar in patients with cancer compared to non-oncological patients. As is already known, we confirmed that IL-6 showed the fastest kinetics of the four biomarkers in all groups (BI, non-BI and SIRS) followed by PCT. The peak value of IL-6 was reached in the first blood sample after the beginning of the febrile episode, decreasing subsequently in the second sample (12–24 h), while the PCT maximum peak was reached at 12–24 h, decreasing later in the third sample (25–48 h). Peak values for MR-proADM were also found in the baseline sample, decreasing during the following h. CRP showed a slower kinetics, reaching its peak value at 36–50 h, which does not make it an ideal marker for patients shortly after the onset of fever. To summarize, when there are no complications and the clinical evolution is adequate, IL-6 and PCT descend rapidly after the peak value [20], as occurred in our series. The CRP biomarker showed similar kinetics but with a 12–24 h delay [21].

A study published in 2004 [22] proved that PCT and IL-6 were more reliable markers than CRP in predicting bacteremia in patients with febrile neutropenic fever. Moreover, IL-6 showed a high negative predictive value on day 1 (close to 89%) to exclude bacteremia/sepsis [23]. Similarly, to these results, we demonstrated that IL-6 was the only biomarker significantly related to the presence of a systemic response and bacterial infection, both quantitatively and qualitatively (cut-off point > 85 pg/mL). Serum biomarkers utility in the initial risk assessment of febrile neutropenia episodes was also reviewed by The Predicting Infectious Complications of Neutropenic Sepsis in Children with Cancer Secretariat [24]. Although most of the studies, including this systematic review, included children with both hematological and solid malignancies, none of them stratified the results according to the underlying diagnosis or the chemotherapy pathway. Furthermore, there are limited data about the kinetics of serum biomarkers in patients without acute infection undergoing different chemotherapy regimens. They concluded that IL-6 and PCT showed proinflammatory and anti-inflammatory responses similar to what is described in non-neutropenic patients [25], which is also in line with the kinetics of IL-6 and PCT observed in our work.

A meta-analysis published in 2015 [26] found that PCT was a specific but less sensitive marker of BI in patients with febrile neutropenia. At the same time, CRP was a sensitive but less specific marker for BI. Thus, a few studies have determined that PCT levels are significantly higher in those patients with systemic disease than those with localized disease [22]. We found that neither PCT nor CRP were useful biomarkers to diagnose BI. To explain this, we need to consider that in patients with BI, the biomarkers increase is mainly due to the acute inflammatory process triggered by infection. Patients in our sample diagnosed with a BI had a rapid recovery with no systemic response observed in the majority of cases. Therefore, CRP and PCT hardly raised their value.

In 2013 Shuaibi et al. showed the usefulness of the determination of MR-proADM and PCT in febrile patients with hematological tumors [27]. The levels of both biomarkers were higher in those with bacteremia than in those without documented infection. Additionally, a study by Debiane et al. [28], demonstrated it to be more useful than CRP in response to therapy. Thus, MR-proADM significantly decreased its levels in responders and increased in non-responders [29], seeming to be related to the degree of organ failure and the severity of the infection. However, in our work, we could not find that MR-proADM provided more data than PCT and IL-6, probably because we did not include any patients with organ failures or very severe infection.

Given these results, biomarkers may provide information about the evolution of the patients and their response to the established treatment. Unlike severe infections with bad evolution, biomarkers do not show a considerable elevation in mild infections with a good outcome [30]. Generally, a single determination of biomarkers is not useful in this context, but several determinations that allow us to establish a kinetics profile are more convenient to predict a good evolution, as was demonstrated in our series.

### 4.2. Study Limitations

This study includes several limitations. Firstly, it was conducted in a single center and the results could be affected by the profile of the patients attending our institution or by the procedures followed in the emergency service of our hospital. Therefore, the results of single-center studies are less generalizable. Secondly, we performed an observational study that does not allow us to draw any conclusions concerning therapeutic interventions. Finally, all the episodes had a favorable evolution and it is well known that oncology patients are warned of the importance of going to the hospital quickly in case of fever, to establish a rapid diagnosis and treatment. Furthermore, the collection of cases during a limited period of time might explain the good outcomes observed in all our patients.

## 5. Conclusions

In our experience, IL-6 kinetics was faster than PCT kinetics and both were faster than CRP in patients with fever and cancer who present a good outcome, as has been previously described in other groups of patients, including patients without cancer. Patients with good evolution showed a rapid increase and decrease of PCT, especially inIL-6 levels.

## Figures and Tables

**Figure 1 children-08-01027-f001:**
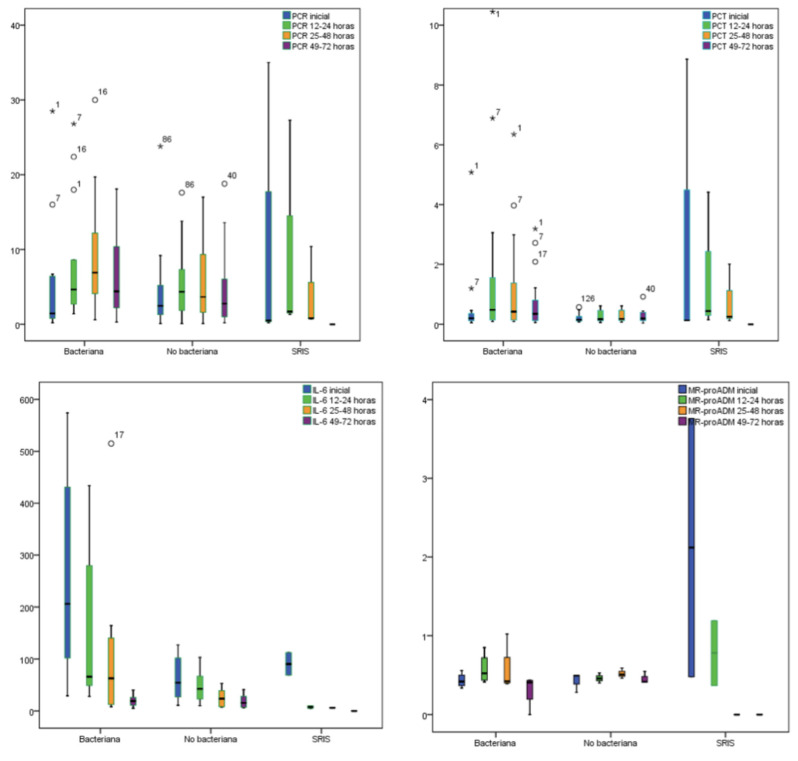
Box plot showing biomarker kinetics. Median and interquartile range are shown in the box. Range from minimum to maximum is represented with whiskers. CRP: C-reactive protein; PCT: procalcitonin; IL-6: interleukin-6; MR-proADM: midregional-pro-adrenomedullin.

**Table 1 children-08-01027-t001:** Demographic and clinical characteristics of patients and episodes.

	PATIENTS (*n* = 37)
Male, *n* (%)	14 (37.8%)
Mean (95% CI)/Median (IQR) in years at the first febril episode	8 (6.4–9.5)/7.3 (8.5)
Solid tumor/HL	19 (51.3%)
ALL/AML/NHL	18 (48.6%)
	**EPISODES (*n* = 134)**
Solid tumor/HL	74 (55.2%)
ALL/AML/NHL	60 (44.7%)
Comorbidities, *n* (%)	86 (64.1%)
No control disease, *n* (%)	23 (17.1%)
Phase of treatment:	
ALL/AML/NHL induction, *n* (%)	19 (14.1%)
Maintenance leukaemia,	26 (19.4%)
BMT, *n* (%)	15 (11.1%)
Solid tumour/HL, *n* (%)	74 (55.2%)
Symptoms	81 (60.4%)
Yes, *n* (%)	
Number of hours previous assistance	
Mean (95% CI)/Median (IQR)	4.4 (3–5.7)/2 (4)
Transfusion (blood +/−platelets)	
Yes, *n* (%)	43 (32.1%)
GCS-F	
Yes, (%)	28 (20.9%)
Final diagnoses:	38 (28.3%)
Bacterial infection, *n* (%)	8 (5.9%)
SIRS, *n* (%)	88 (65.6%)
Non bacterial infection, *n* (%)	

ALL: acute lymphoblastic leukaemia. AML: acute myeloblastic leukaemia. NHL: no Hodgkin’s lymphoma. BMT: bone marrow transplant. HL: Hodgkin’s lymphoma. GCS-F: granulocyte colony-stimulating factor. IQR: intercquartil range. SIRS: Systemic inflammatory response syndrome.

**Table 2 children-08-01027-t002:** Episodes final diagnosis and location of infections.

BACTERIAL INFECTION (*n* = 38)		NO BACTERIAL INFECTION (*n* = 88)		SIRS (*n* = 8)
Bacteriemia	6 (4.4%)	Cutaneous infection	2 (1.4%)	8 (5.9%)
Catheter infection	3 (2.2%)	Fever of unknown origin	36 (26.8%)	
Gastroenteritis	4 (2.9%)	Oropharyngeal candidiasis	2 (1.4%)	
Pneumonia	1 (0.7%)	Upper respiratory infection	48 (35.8%)	
Throat infection	2 (1.4%)			
Urinary tract infection	18 (13.4)			
Sepsis	4 (2.9%)			

N: number of episodes. SIRS: Systemic inflammatory response syndrome.

**Table 3 children-08-01027-t003:** C-Reactive Protein, Procalcitonin, IL-6 and MR-proADM levels in episodes with bacterial infection, systemic inflammatory response syndrome, and no bacterial infection in the four analytical controls.

**Biomarkers levels** **(Baseline)**	**Bacterial infections** **median (IQR)**	**Non bacterial infections** **median (IQR)**	**SIRS** **median (IQR)**	**Value *p*** **median (IQR)**
CRP mg/dL	1.45 (4.7)	1.6 (3.6)	1.4 (5.7)	0.950
PCT ng/mL	0.15 (0.15)	0.13 (0.12)	0.32 (4.49)	0.150
IL-6 pg/mL	* 78 (152.5)	52 (55.5)	^X^ 112 (4143)	<0.005
MR-proADM nmol/L	0.47 (0.16)	0.47 (0.12)	0.75 (2.67)	0.060
**Biomarkers levels** **(12–24 h)**	**Bacterial infections** **median (IQR)**	**Non bacterial infections** **median (IQR)**	**SIRS** **median (IQR)**	**Value *p*** **median (IQR)**
CRP mg/dL	4.45 (5.9)	4.3 (9.9)	12.95 (25.13)	0.842
PCT ng/mL	0.35 (2.14)	0.2 (0.33)	2.43 (14.85)	0.089
IL-6 pg/mL	*^X^ 75.5 (163)	33.5 (51.25)	14 (17)	<0.005
MR-proADM nmol/L	0.46 (0.27)	0.46 (0.19)	0.9 (-)	0.378
**Biomarkers levels** **(25–48 h)**	**Bacterial infections** **median (IQR)**	**Non bacterial infections** **median (IQR)**	**SIRS** **median (IQR)**	**Value *p*** **median (IQR)**
CRP mg/dL	5.20 (8.3)	4,7 (9.5)	3.45 (8.6)	0.360
PCT ng/mL	0.4 (0.74)	0.21 (0.43)	1.13 (8.99)	0.423
IL-6 pg/mL	29 (92)	17 (35.25)	6 (-)	0.540
MR-proADM nmol/L	0.43 (0.34)	0.47 (0.20)	0.93 (-)	0.333
**Biomarkers levels** **(49–72 h)**	**Bacterial infections** **median (IQR)**	**Non bacterial infections** **median (IQR)**	**SIRS** **median (IQR)**	**Value *p*** **median (IQR)**
CRP mg/dL	3.70 (7.20)	3.10 (7.10)	-	0.935
PCT ng/mL	0.31 (0.59)	0.18 (0.27)	-	0.665
IL-6 pg/mL	14 (15)	15.5 (30.75)	-	1
MR-proADM nmol/L	0.41 (0.34)	0.41 (0.21)	-	0.773

CRP: C-reactive protein, PCT: procalcitonin. IL-6: interleukin 6. MR-proADM: midregional-pro-adrenomedullin. SIRS: Systemic inflammatory response syndrome. *: *p* value in the posterior study: *p* < 0.05 with respect to BI-SIRS and BI-NBIh. ^X^: *p* = 0.005 in favor of BI, after excluding SIRS episodes and performing the analysis between BI and NBI.

**Table 4 children-08-01027-t004:** Relation between C-reactive protein, procalcitonin, Interleukin-6 and midregional pro-adrenomedullin levels and number of neutrophils.

Biomarkers Levels (Baseline)	Neutrophils	Median	IQR	Value *p*
CRP (mg/dL)	>1500/μL	1.4	3.20	0.373
	500–1500/μL	2.6	5.20	
	<500/μL	1.9	4.40	
PCT (ng/mL)	>1500/μL	0.12	0.13	0.067
	500–1500/μL	0.20	0.13	
	<500/μL	0.15	0.15	
IL-6 (pg/mL)	>1500/μL	59.5	85.5	0.411
	500–1500/μL	41.5	105.25	
	<500/μL	64	60	
MR-proADM (nmol/L)	>1500/μL	0.48	0.22	0.252
	500–1500/μL	0.52	0.18	
	<500/μL	0.46	0.13	

IQR: interquartile range. CRP:C-reactive protein. PCT: procalcitonin. IL-6: interleukin-6. MR-proAD: midregional-pro-adrenomedullin.

**Table 5 children-08-01027-t005:** Percentage of patients with biomarker levels above their respective cut-off values at the highest level point.

Biomarkers Cutoff Value	Highest Level Point of Each BM	Bacterial Infections	Non Bacterial Infections	SIRS	Value *p*
PCR > 4 mg/dL	25–48 h	73.6%	53.8%	50%	0.343
PCT > 0.5 ng/mL	12–24 h	36.3%	23.2%	50%	0.267
Il-6 > 85 pg/mL	Baseline	47.6%	22.2%	* 71.4%	**0.007**
MR-proADM > 0.5 nmol/L	Baseline	42.1%	31.8%	80%	0.115

CRP: C-reactive protein, PCT: procalcitonin. IL-6: interleukin 6. MR-proADM: midregional-pro-adrenomedullin. SIRS: Systemic inflammatory response syndrome. *: higher values in SIRS, comparing bacterial infections, non bacterial infections and SIRS.

**Table 6 children-08-01027-t006:** Correlation between C-reactive protein, procalcitonin, Interleukin-6 and Midregional-pro-adrenomedullin.

Biomarkers (Baseline)	PCR (Value of *p*)	PCT (Value of *p*)	IL-6 (Value of *p*)	MR-proADM (Value of *p*)
CRP	-	<0.001	0.043	0.261
PCT	<0.001	-	0.045	<0.001
IL-6	0.043	0.045	-	<0.001
MR-proADM	0.261	<0.001	<0.001	-

CRP: C-reactive protein. PCT: procalcitonin. IL-6: interleukin-6. MR-proAD: mid regional-pro-proadrenomedullin. SIRS: systemic inflammatory response syndrome.

## Data Availability

The datasets generated during/or analyzed during the current study are available from the corresponding author on reasonable request.

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
