# Peer review of "Biomarkers and Fever in Children with Cancer: Kinetics and Levels According to Final Diagnosis"

_children, 2021, doi:10.3390/children8111027_

Round 1

Reviewer 1 Report

This is a simple write-up of a study exploring how serum biomarkers change over time in children with fever and neutropenia. It demonstrates consistent answers with the broader literature, with good outcomes linked with resolving raised markers, and speedier responses from IL6 and PCT than CRP.

The paper has the need for a significant copy-editing review. Note the review manuscript offered did NOT have Tables or Figures as part of it and cannot be reviewed.

There are two areas where greater clarity is needed:

  1. Are the categories BI, NBI and SIRS mutually exclusive? This needs describing early on. If mutually exclusive, ANOVA rather than three pairwise comparisons would be more appropriate (but not a deal breaker)
  2. Were these categories determined after multiple analyses, or as part or a pre-determined analysis plan?

Typo 'hints' - NOT exhaustive - follow (based on PDF page and line numbering)

typography of citation; currently as "text1" instead of superscript or "text [1]"
p1 l 39 - nutritional support (not supports)
p1 l 42 - hyphen in evolution un-needed, same in l 43 .. clinical. lots of these
p2 l 50 - "this time point" - better to say 'early' .. 

p2 l 85 - above 16 (ie 17y or older) or "aged 16 years or older"
p4 l 145 - non-Hodgkins (not no Hodgkin)

would prefer not to use BM as biomarkers abbreviation as in paediatric English 'BM' gets used for Bowel Movement or Bone Marrow 

wouldn't usually use men/women for children - either boy/girl or male/female

what stats package was used for the analysis? needs referencing really

Author Response

Dear reviewer,

First of all, thank you for your time invested in correcting this article.   

I would like to respond to all suggestions made:

  1. The paper has the need for a significant copy-editing review. Note the review manuscript offered did NOT have Tables or Figures as part of it and cannot be reviewed.

We have performed copy-editing review. The article presents several tables and figures. We have already written to the editor to report that you did not receive them.

  1. Are the categories BI, NBI and SIRS mutually exclusive? This needs describing early on. If mutually exclusive, ANOVA rather than three pairwise comparisons would be more appropriate (but not a deal breaker)

The categories BI, NBI and SIRS are mutually exclusive. This was described early on in the manuscript.

We did not use ANOVA because quantitative variables (age, hours of fever, place of origin / destination, blood count values ​​and biomarkers) did not behave as a normal distribution (Kolmogorov test, p<0.05), so non-parametric statistics were used in their comparisons (Mann-Whitney U and Kruskal-Wallis test, as needed).

  1. ¿Were these categories determined after multiple analyses, or as part of a pre-determined analysis plan?

These categories were previously determined as part of an analysis plan. During the study, the episodes were classified in one of the three groups according to the final diagnosis. SIRS was defined from the consensus panel of the International Consensus Conference on Pediatric Sepsis (

For the “a posteriori” analysis of the non-parametric tests, Dunn's test and Tukey's test were used.

  1. What stats package was used for the analysis? needs referencing really

The program used was SPSS version 24 (IBM SPSS statitics v24)

The rest of grammatical suggestions have already been changed in the article. The manuscript was also checked by a native English speaker.

Reviewer 2 Report

The manuscript of de Lucio Delgado et al follows several biomarkers of infections over time in children with cancer during fever. As most of these markers normally are not measured simultaneously, this study tries to dissect which biomarkers are best to monitor and during which type of infection.

In its current form it is unclear if and how baseline demographics were changed by discarding these seven patients (ten episodes) and an explanation of why these were removed is lacking. This should be improved textually as well as in Table 1.

The authors must provide statistics more thoroughly. They have indicated p-values into the text for age and sex but these along with tumor types should be detailly included into e.g., table 1, so that readers could see statistics for all factors and interactions. Likewise, statistics and the additional timepoints should be included into Table 3.

Figure 1 cannot be presented in such a way, as at least standard deviations and statistics should be included and methods section on statistical analysis extended.

Also, it would be helpful to provide baseline data of biomarker levels in non-fever conditions.

For ethics reasons the University Hospital name and Ethics Committee approval number should be included.

In general, the entire manuscript should be carefully checked for grammar errors as it seems copied from another template with e.g., many interrupted words.

Author Response

Dear reviewer,

First of all, thank you for your time invested in correcting this article. I would like to respond to the all suggestions made:

  1. In its current form it is unclear if and how baseline demographics were changed by discarding these seven patients (ten episodes) and an explanation of why these were removed is lacking. This should be improved textually as well as in Table 1.

We have challenge the paragraph: “Patients with non-oncological hematological diseases (medullary aplasia, for example) from whom a sample of blood biomarkers had been collected, were excluded for data analysis. Those from whom the first blood sample was collected beyond the first 24 hours after starting the febrile episode were also excluded. The number of discarded episodes was ten (seven patients).”

  1. The authors must provide statistics more thoroughly. They have indicated p-values into the text for age and sex but these along with tumor types should be detailly included into e.g., table 1, so that readers could see statistics for all factors and interactions. Likewise, statistics and the additional timepoints should be included into Table 3.

            The table 1 shows only epidemiological data. This is the reason why we did not put p value (it is only a descriptive table).

  1. Figure 1 cannot be presented in such a way, as at least standard deviations and statistics should be included and methods section on statistical analysis extended.

We tried to include standard deviations in figure 1, but as they are so many values the information provided was confused. We have extended information showed in figure 1 in Table III.  We have changed Table III, Table IV and Table VI because the information was confused. The test has also been modified in order to explain better the results.

  1. Also, it would be helpful to provide baseline data of biomarker levels in non-fever conditions.

We did not collect those data. We only obtained the biomarker levels in fever-condition.

  1. For ethics reasons the University Hospital name and Ethics Committee approval number should be included.

The following sentence has been introduced in the article: Approval from the Autonomic Ethics Committee of Hospital Universitario Central de Asturias (Ethics Committee approval number: 109/15) was obtained”.

  1. In general, the entire manuscript should be carefully checked for grammar errors as it seems copied from another template with e.g., many interrupted words.

We have checked again all the grammar of the article. The manuscript was also checked by a native English speaker.

Round 2

Reviewer 2 Report

The authors have substantially improved text and grammar and the epidemiological argument raised for Table 1 is understandable. However, indications of baseline values could have been added by either, assessing non-fever conditions, use of other patient values from the University Hospital, description of upper and lower detection limits and/or literature. As hardly any of the biomarkers assessed at any of the indicated timepoints is significantly changed it is even more important to openly show data in Figure 1. To me this remains a main weakness of the current study and should be changed by showing all real datapoints or standard deviations. If a graph becomes to confusing perhaps data could be changed into bargraphs or any other type of visualization plots.

Author Response

Dear reviewer,

Thank you again for the careful evaluation of our manuscript.

Following your indications, we have changed figure 1 to box plot to show all the real datapoints including median, interquartile range, range and outliers.

Regarding, biomarker baseline values, we have added data previously published from our University Hospital in patients with non-fever conditions (reference 18)  “C-reactive protein, procalcitonin and interleukin-6 kinetics in pediatric postoperative patients”

Round 3

Reviewer 2 Report

The authors have adjusted the main concern and prepared the manuscript figure more in line with a FAIR Open Data policy.